# Antibody response to SARS-CoV-2 infection in humans: A systematic review

Nathan Post[1‡], Danielle Eddy[2‡], Catherine Huntley[1‡], May C. I. van Schalkwyk[3], Madhumita Shrotri[1,3], David Leeman [2], Samuel Rigby[1], Sarah V. Williams[1], William H. Bermingham [4], Paul Kellam[5], John Maher[6,7], Adrian M. Shields[8], Gayatri Amirthalingam[2], Sharon J. Peacock[2,9], Sharif A. Ismail [2,10,11] *

1 Faculty of Public Health and Policy, London School of Hygiene and Tropical Medicine, London, United Kingdom, 2 National Infection Service, Public Health England, London, United Kingdom, 3 Department of Public Health, Environments and Society, London School of Hygiene and Tropical Medicine, London, United Kingdom, 4 Department of Clinical Immunology, University Hospitals Birmingham, Birmingham, United Kingdom, 5 Department of Infectious Disease, Faculty of Medicine, Imperial College London, London, United Kingdom, 6 School of Cancer and Pharmaceutical Studies, King's College London, London, United Kingdom, 7 Department of Immunology, Eastbourne Hospital, Eastbourne, United Kingdom, 8 Clinical Immunology Service, Institute of Immunology and Immunotherapy, University of Birmingham, Birmingham, United Kingdom, 9 Department of Medicine, University of Cambridge, Cambridge, United Kingdom, 10 Department of Primary Care and Public Health, Imperial College London, London, United Kingdom, 11 Department of Global Health and Development, London School of Hygiene and Tropical Medicine, London, United Kingdom

‡ These authors share first authorship on this work.
* sharif.ismail15@imperial.ac.uk

**Data Availability Statement:** All relevant data are within the manuscript and its Supporting information files.

**Funding:** The authors received no specific funding for this work. MCIvS is funded by a NIHR Doctoral

## Abstract

### Background

Progress in characterising the humoral immune response to Severe Acute Respiratory Syndrome 2 (SARS-CoV-2) has been rapid but areas of uncertainty persist. Assessment of the full range of evidence generated to date to understand the characteristics of the antibody response, its dynamics over time, its determinants and the immunity it confers will have a range of clinical and policy implications for this novel pathogen. This review comprehensively evaluated evidence describing the antibody response to SARS-CoV-2 published from 01/01/2020-26/06/2020.

### Methods

Systematic review. Keyword-structured searches were carried out in MEDLINE, Embase and COVID-19 Primer. Articles were independently screened on title, abstract and full text by two researchers, with arbitration of disagreements. Data were double-extracted into a pre-designed template, and studies critically appraised using a modified version of the Public Health Ontario Meta-tool for Quality Appraisal of Public Health Evidence (MetaQAT) tool, with resolution of disagreements by consensus. Findings were narratively synthesised.

### Results

150 papers were included. Most studies (113 or 75%) were observational in design, were based wholly or primarily on data from hospitalised patients (108, 72%) and had important

Fellowship (Ref NIHR300156). JM acknowledges the support of the National Institute for Health Research (NIHR) Biomedical Research Centre based at Guy's and St Thomas' NHS Foundation Trust and King's College London. SAI is supported by a Wellcome Trust Clinical Research Training Fellowship (Ref No 215654/Z/19/Z). The views expressed in this paper are those of the authors and not necessarily those of the National Health Service (NHS), the NIHR, Public Health England (PHE) or the Department of Health and Social Care.

**Competing interests:** All authors have completed the ICMJE uniform disclosure form at www.icmje.org/coi_disclosure.pdf and declare: no support from any organisation for the submitted work; JM is chief scientific officer, shareholder and scientific founder of Leucid Bio, a spinout company focused on development of cellular therapeutic agents; no other relationships or activities that could appear to have influenced the submitted work. This does not alter our adherence to PLoS ONE policies on sharing data and materials.

methodological limitations. Few considered mild or asymptomatic infection. Antibody dynamics were well described in the acute phase, up to around three months from disease onset, but the picture regarding correlates of the antibody response was inconsistent. IgM was consistently detected before IgG in included studies, peaking at weeks two to five and declining over a further three to five weeks post-symptom onset depending on the patient group; IgG peaked around weeks three to seven post-symptom onset then plateaued, generally persisting for at least eight weeks. Neutralising antibodies were detectable within seven to 15 days following disease onset, with levels increasing until days 14–22 before levelling and then decreasing, but titres were lower in those with asymptomatic or clinically mild disease. Specific and potent neutralising antibodies have been isolated from convalescent plasma. Cross-reactivity but limited cross-neutralisation with other human coronaviridae was reported. Evidence for protective immunity in vivo was limited to small, short-term animal studies, showing promising initial results in the immediate recovery phase.

## Conclusions

Literature on antibody responses to SARS-CoV-2 is of variable quality with considerable heterogeneity of methods, study participants, outcomes measured and assays used. Although acute phase antibody dynamics are well described, longer-term patterns are much less well evidenced. Comprehensive assessment of the role of demographic characteristics and disease severity on antibody responses is needed. Initial findings of low neutralising antibody titres and possible waning of titres over time may have implications for sero-surveillance and disease control policy, although further evidence is needed. The detection of potent neutralising antibodies in convalescent plasma is important in the context of development of therapeutics and vaccines. Due to limitations with the existing evidence base, large, cross-national cohort studies using appropriate statistical analysis and standardised serological assays and clinical classifications should be prioritised.

## Introduction

Severe acute respiratory syndrome coronavirus-2 (SARS-CoV-2), the novel viral pathogen that causes coronavirus disease 2019 (COVID-19) in humans, has spread worldwide since its identification in late 2019. At the time of writing, there have been around 57.9m confirmed cases and 1.4m deaths reported to the WHO [1]. Limited pre-existing immunity is assumed to account for the extraordinary rise in cases worldwide. Characterisation of the human antibody response to SARS-CoV-2 infection is vitally important to inform vaccine development and strategies, and to guide appropriate design, implementation, and interpretation of serological assays for surveillance purposes. Transmission models used to predict the behaviour of the pandemic and plan non-pharmaceutical interventions assume a degree of protective immunity arising from infection with SARS-CoV-2 [2, 3]. A range of clinical and policy interventions to tackle SARS-CoV-2 spread depend on better understanding of the dynamics and determinants of humoral immunity to this virus. These include the proposed use of 'immunity passports', a form of certification for individuals with positive detection of antibodies that can enable them to avoid isolation or quarantine on the assumption they are protected against re-infection [4]; treatment options such as infusion of convalescent plasma or derived immunoglobulin [5];

sero-surveillance to monitor progression of the epidemic in the population [6]; and the nature of the likely response to vaccination and supporting decisions on prioritising use of vaccines.

Experience with other human coronavirus species (HCoV) suggests that partial immunity arises following infection with a variable but generally short (one to two year) duration [7]. Limited data available for the closely related Severe Acute Respiratory Syndrome Coronavirus-1 (SARS-CoV-1) indicate that antibodies able to block viral infection (neutralising antibodies) may persist for up to 17 years following infection [8]. Early clinical studies suggest that the dynamics of antibody response following acute infection with SARS-CoV-2 is similar to other HCoVs. Antibody responses are generally detected against the nucleocapsid (N) or spike (S) proteins, the S1 subunit of which contains the receptor-binding domain (RBD): antibodies against different antigens may have differential dynamics and neutralising effect. The presence of neutralising antibodies (nAb) has been demonstrated in studies of vaccine research and therapeutic use of convalescent plasma [7, 9]. Previous lessons from SARS-CoV-1, Middle Eastern Respiratory Syndrome (MERS-CoV) epidemics and other seasonal human coronaviruses suggest that there is the potential for a decline in population level protection from reinfection over a short period of time, but this is somewhat dependent on initial disease severity [7, 9]. nAbs are likely to be a key metric for protection against infection by viruses such as SARS-CoV-2. However, their dynamics and role in long-term population immunity are not well understood [7]. Furthermore, understanding of the mechanistic correlates of protective immunity in humans remains limited, including the antibody titre and specificity required to confer protection [10].

This is the first of two linked papers reporting results from a systematic review of peer-reviewed and pre-print literature on the immune response to SARS-CoV-2 infection [11]. This paper has three aims. Firstly, to characterise the antibody response to SARS-CoV-2 infection over time and explore the effects of potential correlates of immune activity (including age, time since symptom onset, clinical severity and ethnicity) on the nature of this response. Secondly, to consider relationships between these variables and indirect or relative quantification of antibodies to SARS-CoV-2. Thirdly, to consider the duration of post-infection immunity conferred by the antibody response.

## Materials and methods

This systematic review was carried out according to the Preferred Reporting Items for Systematic Reviews and Meta-Analyses (PRISMA) guidelines. The protocol was pre-registered with PROSPERO (CRD42020192528).

### Patient and public involvement

There was no patient or public involvement in the conceptualisation or design of this review.

### Identification of studies

Keyword-structured searches were performed in MEDLINE, Embase, COVID-19 Primer and the Public Health England library [12] for articles published from 01/01/2020-26/06/2020. A sample search strategy is in **S1 Appendix in** S1 File. Subject area experts were consulted to identify relevant papers not captured through the database searches.

### Definitions, inclusion and exclusion criteria

We included studies in all human and animal populations, and in all settings (laboratory, community and clinical—encompassing primary, secondary and tertiary care centres) relevant to

our research questions. We excluded the following study designs: case reports, commentaries, correspondence pieces or letter responses, consensus statements or guidelines and study protocols.

We focused on studies reporting measured titres (total antibody, IgA, IgG and/or IgM) with follow-up duration of greater than 28 days (which we defined as the limit of the acute phase of illness). Shorter follow-up studies were included if they reported on protective immunity, or immune response correlates. We defined "correlates" as encompassing, among other factors: primary illness severity—proxied by the WHO's distinction between "mild", "moderate", "severe" and "critical" illness [13]; subject age; gender; the presence of intercurrent or co-morbid disease e.g. diabetes, cardiovascular and/or chronic respiratory disease; and ethnicity.

## Selection of studies

Studies were independently screened for inclusion on title, abstract and full text by two members of the research team (working across four pairs), with arbitration of disagreements by one review lead.

## Data extraction, assessment of study quality, and data synthesis

Data were extracted in duplicate from each included study. Extraction was performed directly into a dedicated Excel template (**S2 Appendix in** S1 File). Pre-prints of subsequently published peer reviewed papers were included and results extracted where substantial differences in reported data were identified; if little difference was observed only the peer-reviewed version was retained.

Critical appraisal for each included study was performed in duplicate using a version of the MetaQAT 1.0 tool, adapted for improved applicability to basic science and laboratory-based studies. MetaQAT was selected for its simplicity and versatility in application to studies of all design types [14]. Principal adaptations to the MetaQAT tool are described in **S3 Appendix in** S1 File.

The adapted MetaQAT tool was used to gather both qualitative (narrative) feedback on study quality and scaled responses (yes/no/unclear) for answers to key questions around study reliability, internal and external validity, and applicability, among other fields. These data provided the basis for quality assessments for each paper included in the review.

Study heterogeneity precluded formal meta-analysis. Results were instead synthesised narratively.

## Ethical approval

This was a systematic review based on analysis of openly published secondary data and did not involve humans. No ethical approval was required.

## Results

The PRISMA flowchart for the review is given in Fig 1.

## General characteristics of included studies

150 studies were included, of which 108 (72%) contained data pertaining to antibody response, and 70 (47%) to protective immunity (descriptive statistics for included studies are given in Table 1). The vast majority (108 or 72%) focused on hospitalised patients (i.e. higher severity disease). Eleven studies (7%) considered antibody responses in asymptomatic individuals in the community and only five (3%) investigated protective immunity in this group. Most

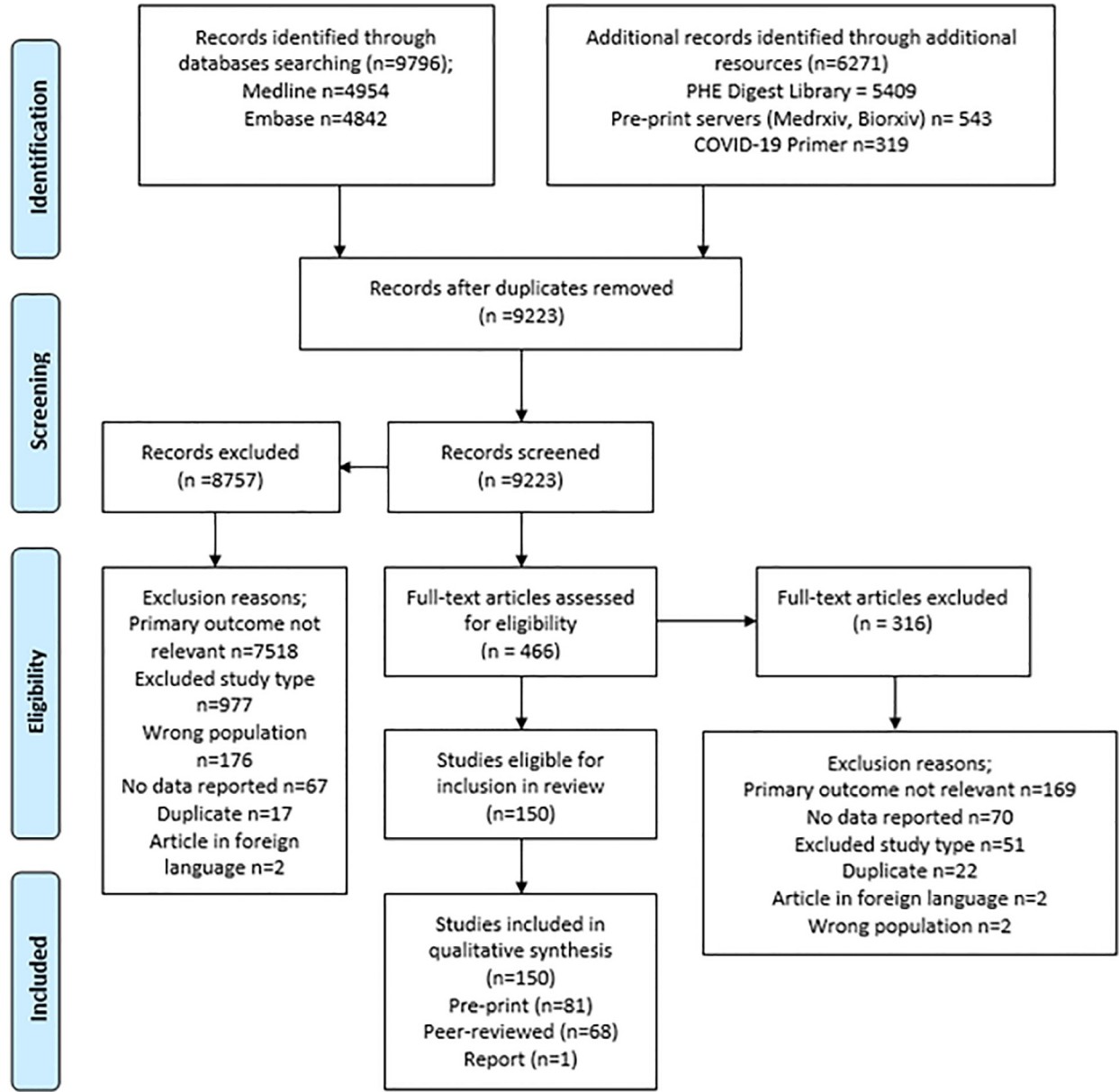

**Fig 1. PRISMA flowchart describing the process of screening and selection of included studies.**

studies had significant methodological limitations. Assays used to detect and quantify antibody response were diverse, with target antigens including spike (S), S1 and S2 subunits, receptor binding domain (RBD) and nucleocapsid (N). Details of assays used, and an overview of strengths and limitations of these is provided in **S4 Appendix in** S1 File.

## Kinetics of the antibody response

**Time to seroconversion.** The majority of individuals in the included studies mounted a SARS-CoV-2-specific antibody response during the acute phase of illness, with many studies reporting 100% seroconversion. Overall seroconversion rates depended on the time point at

**Table 1. Summary of characteristics of included studies.**

| | Antibody response | Protective Immunity | All papers |
|---|---|---|---|
| *Article type* | | | |
| Pre-print | 57 | 43 | 81 |
| Peer reviewed paper | 51 | 26 | 68 |
| Report | 0 | 1 | 1 |
| *Study designs* | | | |
| Cohort | 58 | 18 | 66 |
| Case control | 20 | 14 | 26 |
| Case series | 15 | 8 | 21 |
| Basic science | 4 | 22 | 24 |
| Narrative review | 4 | 2 | 4 |
| Systematic review with meta-analysis | 5 | 2 | 5 |
| Systematic review without meta-analysis | 1 | 1 | 1 |
| Non-randomised trial | 1 | 2 | 3 |
| *Subjects* | | | |
| Human | 104 | 58 | 137 |
| Animal | 1 | 6 | 6 |
| Both | 3 | 6 | 7 |
| *Country of origin* | | | |
| China | 41 | 20 | 54 |
| USA | 15 | 14 | 21 |
| Europe excl UK | 29 | 9 | 35 |
| UK | 7 | 5 | 10 |
| Other countries | 5 | 4 | 7 |
| Multiple populations | 10 | 6 | 11 |
| Lab or animal based* | 1 | 12 | 12 |
| *Study setting* | | | |
| Hospital patients | 70 | 33 | 85 |
| Mixed hospital and community | 18 | 11 | 24 |
| Community | 13 | 5 | 18 |
| Unclear | 6 | 12 | 14 |
| Animal only study | 1 | 9 | 9 |

Note that some studies addressed both Ab and protective responses, and hence are counted in both of the columns relating to these topics above.

which testing was conducted in the disease course, the populations under study, the serology assay platforms used and their specific target proteins. Studies considered time to seropositivity for total antibody and/or individual antibody classes (IgA/IgG/IgM) (Fig 2), although this was often not clearly defined with respect to symptom onset or first positive PCR test. In addition, whilst some studies described specific target proteins of assays used, others were either non-specific or not described. This limited assessment of dynamics of antibodies against specific viral targets, in particular anti-N versus anti-S, the latter of which may be more closely related to protection.

A number of studies reported seroconversion for total antibody (combined IgG, IgM and/or IgA) [15–21], however the focus of findings presented is for specific antibody isotypes. For IgG, mean or median time to seroconversion ranged from 12–15 days post symptom onset [7, 9, 15, 22–26], with wide variation in first to last detection of IgG from four to 73 days post

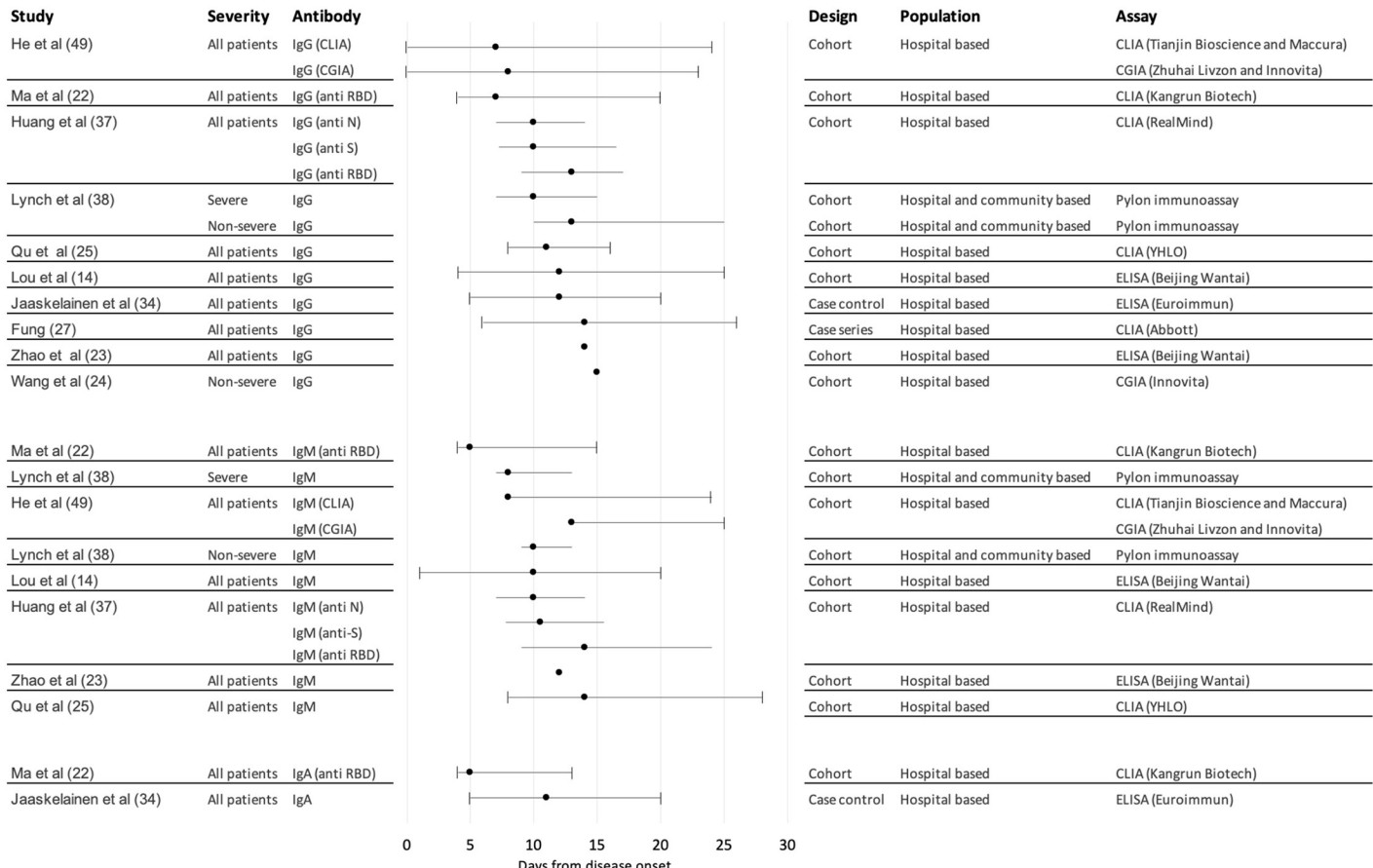

**Fig 2. Forest plot of median time to seroconversion by severity across included studies.** Central points in the forest plot represent the median reported by each study overall. Included studies reported the distribution of times to seroconversion around the median in different ways: lines with whiskers represent ranges (from maximum to minimum around the median); lines without whiskers represent interquartile ranges around the median; for a small number of studies only point estimates were provided.

symptom onset although reporting methods varied by study [15, 27–33]. For IgM, mean or median time to seroconversion ranged from four to 14 days post symptom onset [7, 9, 15, 22–24, 26, 31, 34], again with variations in reporting methods, study quality, and sample size giving rise to uncertainty around findings. Time to seroconversion for IgA was measured in fewer studies, ranging from four to 24 days post symptom onset, although most were within four to 11 days [23, 35, 36], with some outliers, including two reports of 24 days to first detection [37, 38].

**Sequential antibody response.** In line with the expected sequential appearance of antibody isotypes, the majority of studies reported detection of IgM followed by IgG [15, 23, 39, 40]. Nevertheless, this finding was not consistent across all studies. One study measured time to seroconversion for IgA, IgM, and IgG and demonstrated detection of IgA and IgM simultaneously, followed by IgG [23]. One study detected IgG seroconversion in advance of IgM [26], and a study involving African green monkeys reported simultaneous IgM and IgG responses [41]. These disparities may reflect the use of differing antibody assays across a range of species and without standardisation.

**Antibody dynamics over time.** IgG dynamics appeared to follow a pattern of peak, plateau, and persistence at lower levels (Fig 3). After appearance, IgG titres rose to a peak between

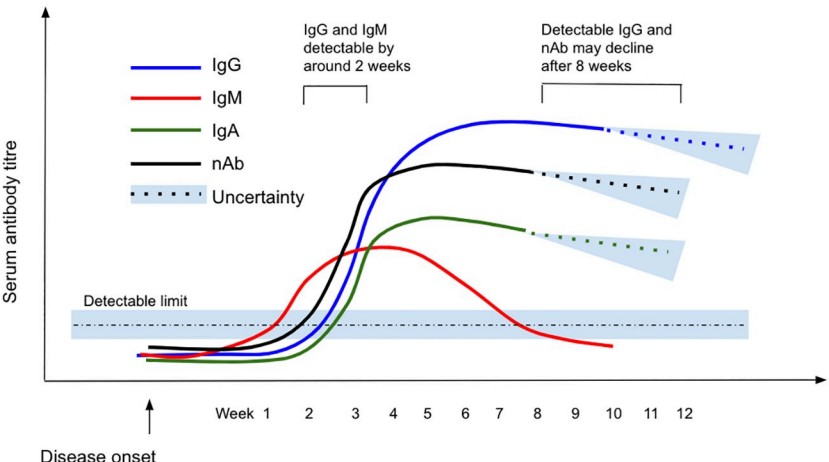

**Fig 3. Schematic showing the scale of IgG/IgM/IgA/Neutralising Ab response over time from disease onset.** Note that the y-axis is illustrative only and therefore no scale is given: this figure gives an indicative overview of findings from all included studies with relative peaks and decline indicated.

three and seven weeks post symptom onset [7, 23, 30, 42–48], with studies recording the presence of IgG in and beyond weeks four [40, 49], five [50], six [23, 51], seven [52, 53], and eight [17, 45, 54–56] post symptom onset. Some studies reported a plateau in virus-specific IgG beyond week three but levels beyond the peak were not well described [32, 57–59]. A decrease in antibody levels was reported in the eighth week post symptom onset by two studies [17, 38], while another reported a decline from the second month after symptom onset [58]. Evidence from a cohort of 40 UK patients suggests a decline in titres after eight weeks [58], although persistence of virus-specific IgG has been described at varying levels up to 12 weeks post symptom onset [43], the longest follow up period among included studies. Dates of last detection were limited by the length of the study follow-up period, rather than confirmation of disappearance of detectable antibody titres.

IgM dynamics follow a 'rise and fall' pattern, with a peak two to five weeks post symptom onset [7, 26, 30, 34, 43, 46, 47, 53, 60] then decline over time to below the detection limit [38, 43, 61]. Beyond the peak, IgM is consistently reported to decrease from as early as two to three weeks [53, 60], to as late as eight weeks [55] post symptom onset, with the majority of studies reporting this decline to occur at between three to five weeks [40, 43, 61, 62]. Virus-specific IgM became undetectable in almost all cases by around six weeks after disease onset in two small but well-conducted cohort studies [53, 63].

Fewer studies describe IgA dynamics compared to IgM or IgG. IgA levels are reported to peak between 16–22 days post symptom onset, although there is no consensus on trends over time [23, 60].

## Correlates of antibody response

Key findings regarding correlates of the antibody response to SARS-CoV-2 infection are summarised in Table 2. Included papers addressed clinical factors (disease severity, co-morbid disease status and symptom profile) and demographic factors (age, sex and ethnicity) although results for many of these factors were conflicting or inconclusive. Across all papers, the definitions of comparator groups were highly variable, including disease severity classifications (severe/mild), outcomes (deceased/mild), and treatment categories (ICU/Non-ICU). The lack of consistency in methods, comparison groups and study design means it is not possible to

**Table 2. Summary of evidence on correlates of antibody response from studies included in this review.**

| Category | Correlate | Dimension or sub-population | Findings |
|---|---|---|---|
| Clinical | Disease severity | Longitudinal trends in Ab production | • Most studies report no relationship between time to seroconversion for IgG or IgM and disease severity [22, 30, 35, 53, 56, 64].<br>• A number of studies report earlier antibody response to more severe disease, including specifically for IgG [7]; a shorter time to peak antibody titre [19, 65–67], and that IgG and IgM persist for longer in severe disease compared to milder cases [68, 69].<br>• Conversely, several studies report an earlier antibody response to milder disease [26, 43], including specifically for IgM [70] and IgA [69, 71]. One study reported shorter time to peak for IgG in mild cases [69], and another reported IgG persisting for longer compared to severe cases [72].<br>• One study reports lower seroconversion rate amongst asymptomatic compared to symptomatic cases [73].<br>• Two other studies on this association were limited by lack of detail on methodology and participant selection [74, 75]. |
| | | Antibody titre | • Ten studies reported that more severe cases have higher IgG titres [23, 34, 39, 42, 62, 76–80], three studies reported no difference between mild and severe cases [44, 52, 81].<br>• Five studies reported severe cases have higher IgM titres [23, 34, 39, 45, 53], two studies reported no difference in between mild and severe cases [63, 81], and three studies reported severe cases have lower IgM titres [80, 82, 83].<br>• Three studies report severe cases have higher IgA titres [23, 69, 78], one study reports no difference between mild and severe cases [82]. |
| | Co-morbid disease | | • Two studies report an association between co-morbidities and seroconversion or antibody positivity [63, 84], with one finding immunocompromised individuals developed a lower response [85].<br>• One study reports antibody responses to be independent of co-morbidities, for both IgA and IgG [78].<br>• One study on this association was limited by sampling bias leading to low quality [86]. |
| | Symptom profile | | • Several studies report an association between COVID-19 symptoms and: seropositivity [87–89], higher titres of IgG [29, 79, 90] and anti-RBD and anti-S antibodies [91].<br>• Two studies reported that asymptomatic healthcare workers did develop antibodies [92, 93].<br>• Fever appears to have a consistent relationship with seropositivity and antibody titres [29, 89, 91], although other symptoms such as ageusia have also been associated [87, 89]. |
| Demographic | Sex | | • Four studies report no association between antibody titres (either IgG or IgM) and sex [21, 78, 94, 95].<br>• One study reports a higher IgG titre in women, in 'severe' patients only [96].<br>• One study reports a delayed peak of antibody response in men [97].<br>• Two studies report a higher proportion of women tested positive for antibodies [87, 95].<br>• One study found higher anti-RBD and anti-S antibodies in male plasma donors [91]. Another study found higher concentrations of IgM in male cases [34]. |
| | Age | Older adults | • Five studies found no association between antibody response and age, for both antibody positivity [63] or IgM/IgG titre [21, 58, 78, 98].<br>• One study found that seroconverters were older than non-seroconverters [84] and another that the concentration of IgG was related to age [34].<br>• Three studies reported that older people had higher titres of IgA and IgG [23], IgM [94], and anti-S and anti-RBD IgG [91]. |
| | | Children | • Two studies found children generally developed a detectable antibody response to SARS-CoV-2 infection [99, 100].<br>• Two studies found children with pneumonia generally mounted lower IgG [101] and IgA responses [102].<br>• One study found no significant difference between the levels of antibody in children and adults [98].<br>• Two studies reported most neonates born to COVID-19 positive mothers had raised IgM [103, 104], and COVID-19 recovered donor breast milk was found to have reactive IgA in one study [105]. |
| | Ethnicity | | • One study reported non-white ethnicity was associated with higher antibody levels than white ethnicity [84]. |

determine whether or how disease severity affects, or is affected by, the antibody response. Most studies showed no association between antibody response and age or sex, and, when taken together, studies that did show associations had inconclusive results and lacked statistical analysis to relate these findings to disease severity. There were virtually no data to describe the immune response according to ethnicity.

## Protective immunity

**Neutralising antibody kinetics.** Across the included studies, the majority of subjects developed detectable neutralising antibodies in response to SARS-CoV-2 infection in both human [7, 18, 22, 29, 76, 100, 106–127] and animal [41, 128–131] participants. However, neutralising antibody titres were low in a substantial minority of participants. A small although relatively robust cohort study found almost all participants (94%, n = 19) generated an antibody response capable of neutralising 42–99% of pseudovirus in a carefully validated assay 14 days after exposure [124]. Another well conducted cohort study also found most patients (91%, n = 22) developed a neutralising antibody response by 21 days after disease onset [125]. However only three quarters developed titres over 1:80. A larger case-control study including a sample of largely non-hospitalised convalescent patients demonstrated most participants (79%, n = 149) had low neutralising antibody titres (<1:1000) after an average of 39 days following disease onset, while only 3% showed titres >1:5000 [127]. Notably, RBD-specific antibodies with potent antiviral activity were found in all individuals tested, suggesting specific neutralising antibodies are produced following infection despite low overall plasma neutralising ability [127].

Neutralising antibodies were generally detectable between seven to 15 days following disease onset [7, 18, 76, 106, 125, 126, 132, 133], increasing over days 14–22 before plateauing [22, 106, 110, 111, 132, 133] and declining over a period of six weeks [106, 111, 126, 134]. Evidence from one pre-print study suggests neutralising antibody titres reduced significantly among 27 convalescent patients around six weeks following disease onset to a mean neutralisation half maximum inhibitory dilution (ID50) of 596 [51] although this study was at risk of bias due to minimal reporting on methods used for participant selection. A second preprint study, also limited by lack of reporting on inclusion criteria, found neutralising antibodies became undetectable in four of 11 previously detectable cases [126]. Further high-quality evidence is required to fully evaluate the apparent waning of the neutralising antibody response over time. To date no studies have determined neutralising titres in upper respiratory tract samples.

**Correlates of neutralising antibody production.** Clinical and demographic correlates of the neutralising antibody response are described in Table 3. Neutralising antibody responses correlated with disease severity in all studies in which this association was tested [7, 43, 49, 108, 117, 126, 127, 135–137]. Importantly, the few studies that investigated asymptomatic cases found those individuals were considerably less likely to develop detectable serum neutralising antibody responses than cases with symptoms. With regard to age and sex, evidence was mixed and a limitation across all papers was a lack of statistical adjustment for severity.

**Correlation of neutralisation with specific antibodies.** The level of neutralisation was found to correlate with a wide range of specific antibodies. Most studies suggested that neutralisation ability broadly correlated with total virus-specific IgG [29, 49, 109, 116, 127, 138–140]. Specifically, a number of well-conducted studies found that neutralisation ability correlated positively with anti-S IgG [49, 114, 127, 138] or anti-RBD IgG [81, 114, 116, 127]. There was more limited evidence for correlation with anti-RBD IgM and IgA in peer reviewed studies with appropriate statistical analyes [51, 125, 132, 138].

**Table 3. Summary of evidence on correlates of neutralising antibody response from studies included in this review.**

| Category | Correlate | Dimension or sub-population | Findings |
|---|---|---|---|
| Clinical | Disease severity | Longitudinal trends in Ab production | • One study reported asymptomatic cases with neutralising antibodies were more likely to lose detectable neutralising antibodies in the convalescent phase [90]. |
| | | Antibody titre | • Ten studies reported a higher titre of neutralising antibodies in more clinically severe cases [7, 43, 49, 108, 126, 127, 134–137].<br>• One study reported undetectable neutralising activity in plasma from a majority of asymptomatic cases [126]. |
| Demographic | Sex | N/A | • Five studies reported neutralising antibody response was positively correlated with male sex [108, 110, 127, 135], although it is unclear how this is related to disease severity [49].<br>• One study reported a positive correlation between neutralising antibody formation and female sex [126]. |
| | Age | Older adults | • Two studies reported increasing neutralising antibody response with increasing age [20, 133].<br>• Two studies reported no association between neutralising antibodies and age [110, 126]. |
| | | Children | • Two studies reported children can develop a neutralising antibody response [48, 100]. |
| | Ethnicity | N/A | • One study reported that individuals with Hispanic/Latino ethnicity were more likely to have detectable neutralising antibody responses (although this study was based on a convenience sample in an atypical cohort—US service personnel on a warship) [107]. |

A number of basic science studies also identified specific neutralising antibodies. The majority of these studies were of variable but moderate quality, and heterogeneity between assays limits comparability of findings. A well-conducted basic science study by Rogers *et al* highlighted the important role of RBD binding antibodies in neutralisation in a pseudovirus assay, with findings supported by an effective animal re-challenge model [141, 142]. This study also reported that SARS-CoV-2 infection elicited a strong response against the S protein. However, few of these antibodies were neutralising, in agreement with other results [143, 144]. RBD-specific antibodies were also shown to have potent neutralising activity in a range of other small studies [112, 143, 145–151], including one using an IgA isotype [152]. Neutralising ability correlated in particular with competition for the angiotensin converting enzyme-2 (ACE2) receptor [112, 114, 145]. Two studies demonstrated a lack of association with affinity [115, 145], although a moderate correlation with binding affinity was reported in one study [146]. Potently neutralising N specific antibodies were isolated in other studies [115, 148], and the potential for antibodies binding to protease cleavage sites as alternatives to RBD isolated from convalescent plasma has also been identified [153], suggesting an important role in preventing antibody dependent enhancement of viral entry.

Few studies investigated B cell responses in detail. A study by Galson *et al* of 19 hospitalised patients demonstrated clonal expansion and induction of a B cell memory response (possibly to other circulating coronaviridae) but that the predominant expansion was in the naïve B cell population [154]. Strong convergence of response emerged across different participants, which was judged to be associated with disease severity, and these findings were consistent with a large and well conducted case control study by Robbiani *et al* [127].

**Correlation of antibodies with viral load.** Several studies investigated the relationship between SARS-CoV-2-specific IgG and viral load [155, 156] or the co-existence of antibodies and viral RNA [15, 24, 25, 38, 42, 46, 61, 63]. In a large cohort study, the presence of SARS-CoV-2 anti-N IgG was significantly correlated with reduced viral load (measured as cycle threshold (Ct) >22, which was also associated with lower mortality) [155]. This was consistent with a study by To *et al* which correlated increasing anti-N IgG titres with decreasing median viral load from 6.7

to 4.9 log10 copies per mL between weeks one to three [156]. Another relatively large cohort study including controls had similar findings but did not quantify viral load [24]. Together these findings suggest the persistence of detectable RNA despite clinical recovery, and although viral loads generally reduced in the convalescent phase, co-existence of SARS-CoV-2 specific IgG and detectable SARS-CoV-2 RNA could be identified in a small number of patients for up to 50 days following seroconversion [25]. Other studies were mixed, with one finding higher levels of specific antibodies correlated with viral clearance within 22 days [38], and another finding weaker IgG response correlated with viral clearance within seven days after antibodies become detectable [42], although both of these findings are subject to a number of limitations. Importantly, one included study attempted to associate re-detection of viral RNA with the presence of specific antibodies, finding that IgG titres began to decrease immediately following recovery although this was not associated with whether RNA was re-detected. Across all included studies, high quality evidence for re-infection or lasting immunity was lacking.

**Re-exposure to SARS-CoV-2.**   Studies exploring re-exposure to SARS-CoV-2 virus were limited to seven animal studies of variable quality. Broadly, two areas were explored; exposure following a primary infection with SARS-CoV-2 [128–130, 157] and re-exposure following passive transfer of neutralising antibodies [130, 141, 142].

Following primary infection, timing of re-challenge varied between 20–43 days post inoculation. All studies but one [129] demonstrated some level of protection from reinfection with a study in nine macaques showing a significant reduction in viral titres (p<0.001) and reduced clinical symptoms [128]. Similar findings were reported in a hamster model [131, 157]. In a smaller ferret study, clinical findings following reinfection were mixed with the re-challenged group demonstrating increased weight loss compared to naive ferrets. However, the authors acknowledged that the sample size (n = 4) was too small to draw wider inference [129].

Two studies examined protection from SARS-CoV-2 infection following the passive transfer of neutralising antibodies in Syrian hamster models [130, 141, 142]. Following transfer of highly potent neutralising antibodies 12 hours prior to infection, hamsters showed lower viral titres and fewer clinical symptoms of COVID-19. However, following transfer of less potent neutralising antibodies, one to two days prior to infectious challenge, results were mixed demonstrating their inability to fully neutralise the virus [141]. Data on protection from re-infection in humans was not identified in the included papers, therefore conclusions on protective immunity are limited.

**Cross-reactivity with other coronaviruses.**   There is limited evidence on the cross-reactivity of antibodies specific to other coronaviruses [8, 49, 51, 133, 158, 159]. Using a variety of assays, several in-vitro studies explored both cross-reactive antibody-binding responses and cross-neutralisation between SARS-CoV-2 and seasonal HCoVs, MERS-CoV and SARS-CoV-1. Cross-reactive antibody-binding responses appear to be highest between SARS-CoV-1 and SARS-CoV-2, however cross-neutralisation is rare and where reported is weak [49, 133, 158]. Whilst seasonal HCoVs are more common in the population, only 10% of sera exposed to HCoVs demonstrated cross-reactivity again with very little neutralisation activity [159]. A study comparing cross reactivity in children and older participants found children had elevated CoV-specific IgM compared to more mature class-switched specific IgA and IgG [160]. All studies were performed in-vitro and recognised the need for in-vivo investigation.

## Discussion

### Summary of findings

This review narratively synthesis the findings from 150 studies published by the end of June 2020. The pace of production of evidence over time in regard to SARS-CoV-2 has been

exceptional, and further evidence is now available with which to contextualise these earlier contributions to our understanding of SARS-CoV-2 infection. In the discussion that follows we summarise key findings from the included studies and highlight in addition (where relevant) evidence published since our searches were completed.

Most people who experience symptomatic SARS-CoV-2 infection undergo seroconversion to produce a detectable, specific antibody response in the acute phase (≤28 days). The kinetics of the antibody response to SARS-CoV-2 follow typical immunological paradigms: virus-specific IgM rises in the acute phase to a peak around two to five weeks following disease onset, then declines over a further three to five weeks before becoming undetectable in many cases; IgG peaks later (three to seven weeks following disease onset), then plateaus, persisting for at least eight weeks with some evidence suggesting a moderate decline over that period. However, understanding of IgG dynamics over time is limited by the understandably short duration of follow up in studies published for inclusion in this review. Studies with longer follow are now be starting to show persistence of IgG for at least three to four months [161, 162].

Evidence suggests the majority of those infected with SARS-CoV-2 develop nAbs—a finding consistent with that for SARS-CoV-1 and MERS-CoV [7]. The magnitude of this response appears to correlate with disease severity, although not necessarily the kinetics. Neutralising antibodies are initially detectable from around seven to ten days, peaking at around three weeks and then declining. Further evidence is required to evaluate comprehensively the apparent waning of the nAb response over time: in the studies included here although nAb may be detectable, higher quality studies suggest that titres are generally low, and the response is short lived. Although this is supported by recently published data (beyond the date cut off for inclusion in this study) from a UK cohort of healthcare professionals with follow up to three months [163], durable neutralising antibody response up to seven months have now been described [164]. Ongoing vaccine research has highlighted a need for evidence of longer-term protection due to nAbs, and the titres at which these effects are achieved—neither of which were fully addressed by studies included in this review. A number of potent, specific nAbs have been identified—in line with findings for other HCoVs. This is particularly the case for neutralising anti-RBD antibodies [165], and is consistent with data emerging from vaccine development studies showing that protective antibodies can be induced [166–168].

Data on correlates of the antibody response in this review were incomplete, inconsistent or contradictory. It is not possible to draw robust conclusions on the associations of antibody response with age, sex, ethnicity or comorbidities, and although disease severity positively correlated with higher IgG antibody titres in a number of studies, distinguishing causation from correlation is not possible. More recent evidence shows that while responses may be of lower magnitude in milder cases, they are still elicited and may be protective—although evidence on response patterns in milder illness remains in short supply [164, 169].

The size of the detectable nAb response appears to be associated with male sex (although the effect of disease severity was not controlled for); this is a surprising finding given the now well-recognised association between male sex and poor COVID-19 outcomes [170]. A recent study has characterised sex differences in the cytokine and cellular response, although identified no differences in anti-S1 IgG or -IgM between male and female cases [171]. With regard to age, recent data published beyond the cut off for this review has shown distinct antibody responses in children and adults, which may inform future understanding of the disease course in different age groups [172]. This is a welcome addition to the literature considering the lack of studies on children included identified for our review.

Available data on protection following primary infection for this review were limited to small scale animal models which consider re-exposure rather than reinfection. Primary infection appears to provide a degree of protection to reinfection up to day 43 post primary

inoculation but no further data were available at later time points. The success of passive transfer of nAbs for protection against SARS-CoV-2 infection appears to be dose dependent, although no data exist around the importance of affinity, isotype or immunoglobulin subclass. Given the possible reduction in nAb titre over time the protection they provide could be limited. However, more recent evidence from a large cohort of UK health professionals has shown anti-S IgG generated following natural infection may protect from re-infection up to six months [173], consistent with emerging data on the durability of neutralising antibodies. Further evidence on longer durability and correlates is required.

Some emerging evidence has suggested pre-existing humoral immunity arising from infection with seasonal HCoVs, although these data remains difficult to interpret [159]. Overall, our review found limited evidence of cross reactivity between SARS-CoV-2 and other HCoVs, but cross-neutralisation is rare and when it does occur, fails to fully neutralise the SARS-CoV-2 virus.

## Strengths and limitations

This study presents, to our knowledge, the first comprehensive overview and critical appraisal of studies investigating the antibody response to SARS-CoV-2 infection over the first 6 months of the pandemic. The results presented here have a range of implications for treatment and policy, as well as providing a useful basis for building further research. Our findings are nevertheless limited both by aspects of the review methodology and by shortcomings in the included literature. The comprehensiveness of systematic reviews is always dependent on search strategy, and some results relevant to the research question may have been missed; as with all systematic reviews, our findings cannot account for unpublished negative results; and importantly, the pace of evidence production on SARS-CoV-2 means that systematic review research is inherently at risk of missing new, divergent data.

Limitations of the underlying evidence base were considerable. A majority of included studies were of variable but generally moderate quality. Study populations were highly variable, as were the assays used, along with the rigour with which they were described, verified and validated against their target populations. There are efforts in the UK to standardise laboratory SARS-CoV-2 assays use through the National External Quality Assessment Service (NEQAS), but these are early stage and no comparable international initiatives yet exist to support comparability of research findings. Longitudinal follow-up for durations greater than 50–60 days was rare, although we note this was limited by the timeline of the review for this novel pathogen. Many studies did not perform statistical analysis of findings; in particular, studies of putative correlates of immune response usually failed to control for the effects of potential confounders. Small sample sizes were common, as were study populations selected by convenience which, although common for clinical cohort studies, are prone to bias. Additionally, a large body of the evidence drew from pre-print publications which have not been subject to peer-review. While efforts were made to account for this during synthesis and reporting, reporting standards in these publications were highly variable and there is no validated system at this time for weighting evidence from pre-print publications relative to peer-reviewed papers.

Finally, a substantial proportion of included studies failed to appropriately make statements on ethical approval for studies or the use of consent for participation. There were two underlying factors behind this. First, papers uploaded to pre-print servers routinely failed to include an ethical statement: although this may be updated by the time of the peer reviewed publication, the wide use of pre-print material in the current phase of the pandemic suggests ethical statements should be routinely made in these uploaded manuscripts. Second, a number of studies where ethical statements were made reported that informed consent for participation

was waived due to the pandemic situation: such measures limit the confidence with which such research can be used, particularly in an academic field where the use of informed consent should be routine, and where the ethical conduct of studies with human subjects should be the norm.

## Implications for policy

We identify two main policy implications arising from this work. At individual level, continuing uncertainty concerning the nature of the humoral response to SARS-CoV-2 makes it difficult to determine what the practical meaning of serologically-detected antibody response is with respect to sterilising immunity. Short follow-up periods, as well as the use of binary (positive/negative) serological tests in many studies continue to limit what can be said about the granularity of the immune response over time—and by implication, how best to interpret the results of serological testing with respect to individual susceptibility to infection [174]. We did not identify any studies considering risk of re-infection with SARS-CoV-2, which might provide an alternative perspective on susceptibility to infection. Further studies published following the completion of this review shed additional light in this area, although this question remains under active investigation.

At a population-level, important policy implications arising from these data on antibody response relate to both surveillance and control. Serological tests have been evaluated predominantly in acutely unwell, hospitalised patients (without appropriate validation against mild disease or in people with asymptomatic infection) and seroprevalence estimates from this work should therefore be treated with caution. A recent Cochrane review emphasises the risk of false-positive and false-negative results under different population prevalence scenarios [175]. However, in the UK, nationally validated assays have been evaluated with convalescent samples from community participants and a number of large-scale sero-surveys now use these [176–178]. Clear understanding of the kinetics of the response, particularly for the specific N and S antigens, is important for the interpretation of seroprevalence studies. Serological tests remain variable in performance, and a major constraint in interpreting findings across different studies.

With regard to control, the evidence here for lasting protective immunity, or lack thereof, post-infection, may suggest it is too early to recommend the use of 'immunity passports'. A range of promising data have been identified to support further investigation of treatment with convalescent plasma or immunoglobulin. The basic science underlying the antibody/virus/host cell interaction is starting to be described, with promising findings related to vaccine development: most recent vaccine data shows they are able to generate robust humoral responses [166]. For vaccines strategies for implementation will also require a thorough understanding of the likely impact in different population groups. Initial findings presented here give useful context to this, although further research is needed.

## Onward research questions

The limited amount of data on antibody dynamics for mild and asymptomatic cases, which are likely to make up a significant proportion of infections, is a particularly important gap in the literature that will need to be addressed to improve understanding and definition of the varied clinical phenotypes associated with SARS-CoV-2 infection, although progress is starting to be made in this area. Investigating the relationship between antibody response and correlates including age, sex, ethnicity and disease severity through high-quality, large-sample studies using well validated assays and incorporating appropriate statistical testing of results should be prioritised.

Mutations affecting infectivity and potential resistance to vaccines are an emerging threat. There has been considerable recent attention on the likely impact of mutations to the S protein arising from infection in non-human hosts transmitting back to humans. A range of mutations, and their resistance to neutralising monoclonal antibodies, have been characterised [179]; however, further work in the area will be essential for understanding the roll out of vaccines in populations around the world.

Evidence on immunity beyond six months following primary infection or vaccination is urgently needed. Evidence of immunity following vaccination is being explored through various vaccine trials (e.g. ChAdOx1 nCoV-19) [166]. However, further longitudinal studies of those already infected with SARS-CoV-2 is required to examine the degree of protection arising from prior infection.

## Conclusion

Studies on the immune response to SARS-CoV-2 is of variable quality, and comparison of findings is difficult. A longer-term view and a more comprehensive assessment of the role of demographic characteristics and disease severity is required. Larger, high-quality, longitudinal studies, with appropriate statistical analysis, consistent use of established and well-validated serological assays matched to clearly defined clinical phenotypes should be prioritised.

## Supporting information

**S1 File.**
(DOCX)

**S2 File. PRISMA checklist.**
(DOCX)

## Acknowledgments

We thank Professor Mike Ferguson from the School of Life Sciences, University of Dundee, for comments on the research questions and initial outputs from this work; and Professor Mark Petticrew from the Faculty of Public Health and Policy, London School of Hygiene and Tropical Medicine for advice on methodological aspects of this study. We are also grateful to Anh Tran (Senior Knowledge and Evidence Manager), Nicola Pearce-Smith (Senior Information Scientist), Paul Rudd (Knowledge and Evidence Specialist—COVID-19) and James Robinson (Knowledge and Evidence Specialist—North) from Public Health England's Knowledge and Library Services for support in conducting the literature searches on which this review was based.

## Author Contributions

**Conceptualization:** Danielle Eddy, May C. I. van Schalkwyk, David Leeman, Paul Kellam, Gayatri Amirthalingam, Sharon J. Peacock, Sharif A. Ismail.

**Investigation:** Nathan Post, Danielle Eddy, Catherine Huntley, May C. I. van Schalkwyk, Madhumita Shrotri, David Leeman, Samuel Rigby, Sarah V. Williams, Sharif A. Ismail.

**Methodology:** May C. I. van Schalkwyk, Sharif A. Ismail.

**Project administration:** Danielle Eddy, Sharif A. Ismail.

**Supervision:** Gayatri Amirthalingam, Sharon J. Peacock.

**Validation:** William H. Bermingham, Paul Kellam, John Maher, Adrian M. Shields, Gayatri Amirthalingam, Sharon J. Peacock.

**Writing – original draft:** Nathan Post, Danielle Eddy, Catherine Huntley, May C. I. van Schalkwyk, Madhumita Shrotri, David Leeman, Sharif A. Ismail.

**Writing – review & editing:** Nathan Post, Danielle Eddy, Catherine Huntley, May C. I. van Schalkwyk, Madhumita Shrotri, David Leeman, Samuel Rigby, Sarah V. Williams, William H. Bermingham, Paul Kellam, John Maher, Adrian M. Shields, Gayatri Amirthalingam, Sharon J. Peacock, Sharif A. Ismail.

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
