## [Decision Letter · Decision Letter 0]

11 Nov 2020

PONE-D-20-27927

Antibody response to SARS-CoV-2 infection in humans: a systematic review

PLOS ONE

Dear Dr. Ismail,

Thank you for submitting your manuscript to PLOS ONE. After careful consideration, we feel that it has merit but does not fully meet PLOS ONE’s publication criteria as it currently stands. Therefore, we invite you to submit a revised version of the manuscript that addresses the points raised during the review process.

First, let me apologize for the delay in the review process. The two reviewers had widely differing opinions regarding the merit of the review article (reject; minor revision). My attempt to get a third reviewer was unsuccessful (at least in a timely manner).

Personally, I lean more towards the minor revision status, but did not want to discount Reviewer 1's comments about the importance of updating the citations and avoid judging them on quality before they have undergone pre-peer review.  Please update and adjust the review accordingly, provide a rebuttal letter and we can advance the manuscript upon re-submission. 

We look forward to receiving your revised manuscript.

Kind regards,

Nicholas J Mantis

Academic Editor

PLOS ONE

Journal Requirements:

"All authors have read the journal's policy and declare: no support from any organisation for the submitted work; JM is chief scientific officer, shareholder and scientific founder of Leucid Bio, a spinout company focused on development of cellular therapeutic agents; no other relationships or activities that could appear to have influenced the submitted work."

Reviewers' comments:

Reviewer's Responses to Questions

**Comments to the Author**

1. Is the manuscript technically sound, and do the data support the conclusions?

Reviewer #1: No

Reviewer #2: Yes

2. Has the statistical analysis been performed appropriately and rigorously? 

Reviewer #1: N/A

Reviewer #2: Yes

3. Have the authors made all data underlying the findings in their manuscript fully available?

Reviewer #1: Yes

Reviewer #2: Yes

4. Is the manuscript presented in an intelligible fashion and written in standard English?

Reviewer #1: Yes

Reviewer #2: Yes

5. Review Comments to the Author

Reviewer #1: In this outdated review, Ismail and colleagues provide a systematic review on antibody responses to SARS-CoV-2 infection in humans. Such an endeavour would be useful if up to date. It is odd to receive a review in October that covers manuscripts published no later than 26/06/2020, particularly when one of the areas that it aims to cover is antibody kinetics.

A major problem of this work is how the authors grade studies quality. Are the authors experts in all what they have reviewed? If they are not, then they should refrain from attempting such evaluation.

It is unclear whether the "adapted MetaQAT tool" robust enough to report on quality on all the subjects discussed in this review? Authors should refrain from grading the studies unless they are truly experts in the field and explain why they grade a study as low, medium or high. Several studies referenced by the authors have been updated since first publication in preprint, have been quite improved and are now published in top peer-reviewed journal. I was truly surprised to read that some peer-reviewed articles, published in top journals are graded as medium or low by the authors of this review. Do the authors, who cannot possibly be experts in all the areas covered in this review, believe that they know better than reviewers from these top journals?

Yet, the authors indicate that “most studies were of moderate quality”…

Table 2 is incomplete at best and clearly indicates that the authors have not read in detail the articles they reference. I started completing the table to indicate where the right reference should go but it is not the role of this reviewer to perform such task. This table should be completed once the authors read the references in detail (not only the abstracts).

Some of the research questions raised by the reviewers are being answered with very good articles published lately and unfortunately not included in this review. While it is clear that as soon as someone starts to write a review on SARS-CoV-2 it becomes immediately outdated, the current review should include work at least until October.

Reviewer #2: The manuscript submitted by Post et. al. is a systematic review of 150 serological studies on the humoral response to SARS-CoV-2 infection. The authors attempt to determine a consensus of the SARS-CoV-2 antibody response over time, the relationship of antibody responses to immune correlates, and the duration of humoral immunity. Importantly, the authors highlight the fact that discrepant results between serological studies are likely derived from the use of different assay platforms, antigens used, and patient cohort stratification. Their meta-analysis was able to predict preliminary kinetics for SARS-CoV-2 specific IgM, IgA, IgG, and neutralizing antibody. However, no consensus was obtained for immune correlates of the antibody response. Overall, the authors present a well-organized meta-analysis of the current serological studies on COVID-19, and I recommend this manuscript for publication with a minor edit as follows:

1. In Figure 3, please indicate which antigen/s were used for the kinetic prediction. If aggregate data was used from multiple antigens, be sure to clearly describe in the text and figure legend.

6. PLOS authors have the option to publish the peer review history of their article (what does this mean?). If published, this will include your full peer review and any attached files.

Reviewer #1: No

Reviewer #2: No

---

## [Author Response · Author response to Decision Letter 0]

2 Dec 2020

Department of Primary Care and Public Health

Imperial College London, UK

FAO Nicholas J Mantis

Academic Editor

PLoS ONE

01 December 2020

Dear Professor Mantis

Thank you for your email of 11/11/2020 providing reviewer comments on our manuscript “Antibody response to SARS-CoV-2 infection in humans: a systematic review”. We have now reviewed these and outline point-by-point responses below. 

Editorial comments

"All authors have read the journal's policy and declare: no support from any organisation for the submitted work; JM is chief scientific officer, shareholder and scientific founder of Leucid Bio, a spinout company focused on development of cellular therapeutic agents; no other relationships or activities that could appear to have influenced the submitted work."

RESPONSE: Thank you – we have now included a statement as outlined above in lines 635-6 in the manuscript. 

RESPONSE: We have now addressed this. The ethical approval statement has been moved to the methods section only, in lines 186-88.

RESPONSE: We have now addressed this.

Reviewer #1 

COMMENT: In this outdated review, Ismail and colleagues provide a systematic review on antibody responses to SARS-CoV-2 infection in humans. Such an endeavour would be useful if up to date. It is odd to receive a review in October that covers manuscripts published no later than 26/06/2020, particularly when one of the areas that it aims to cover is antibody kinetics.

Some of the research questions raised by the reviewers are being answered with very good articles published lately and unfortunately not included in this review. While it is clear that as soon as someone starts to write a review on SARS-CoV-2 it becomes immediately outdated, the current review should include work at least until October.

RESPONSE: Thank you for this comment. As the reviewer notes above, in the very fast-moving world of SARS-CoV-2 research at this time it is impossible to produce a manuscript that is fully up to date at any given time point. In addition, the comprehensive nature of the approach to article screening, selection, quality appraisal, data extraction and synthesis in systematic reviews means that the process of collating data after searches are performed is time consuming – the rigour of the approach is the primary means of reducing bias in a systematic review. We submitted this manuscript to PLoS ONE at the beginning of September 2020 – i.e. just over 2 months after the searches were completed – an unusually short time after the searches were completed by comparison with a majority of systematic reviews in the published literature. The Cochrane Collaboration suggests an average timeline of 12 months for systematic reviews from inception to submission; this review was completed in around 4 months, reflecting the context in which the work was undertaken and our goal of synthesising rapidly emerging evidence produced in response to a global pandemic.

We do recognise the reviewer’s concerns around how up-to-date the manuscript is, and for this reason have added material in the discussion to reference more recently published literature (see the “Summary of Findings” section) up to 22/11/2020 drawing on expert input from members of the review team, and also relevant material captured in the Public Health England (PHE) COVID-19 literature digest, an evidence tracking tool produced by the PHE Knowledge and Library Services, and which is publicly available at: https://phelibrary.koha-ptfs.co.uk/coronavirusinformation/#DailyEvidenceDigest. However, we were unable because of capacity constraints to run searches in full to cover the period between 26/06/2020 and the present – and would have faced similar issues over the timeliness of the data if we had done so, given the need to screen, select, critically appraise and data extract from these sources.

COMMENT: A major problem of this work is how the authors grade studies quality. Are the authors experts in all what they have reviewed? If they are not, then they should refrain from attempting such evaluation.

It is unclear whether the "adapted MetaQAT tool" robust enough to report on quality on all the subjects discussed in this review? Authors should refrain from grading the studies unless they are truly experts in the field and explain why they grade a study as low, medium or high. Several studies referenced by the authors have been updated since first publication in preprint, have been quite improved and are now published in top peer-reviewed journal. I was truly surprised to read that some peer-reviewed articles, published in top journals are graded as medium or low by the authors of this review. Do the authors, who cannot possibly be experts in all the areas covered in this review, believe that they know better than reviewers from these top journals?

Yet, the authors indicate that “most studies were of moderate quality”…

RESPONSE: Thank you for this comment. Systematic review teams typically comprise a mixture of subject area specialists and those with expertise in review methodologies – it is highly unusual, and indeed usually impossible, for systematic review teams’ expertise to span all subject areas covered by reviews which tend by their nature to be broad. Our review team similarly comprises immunologists, microbiologists, epidemiologists, public health specialists and those with methodological expertise in systematic reviews. This mixed team structure was important in view of the multi-faceted nature of the review and need to consider basic science, clinical, public health and policy implications of the research reviewed. In addition, the structured approach to the review – clarity on search protocols, documented procedures for screening and selection of studies, data extraction from them and quality appraisal – is specifically designed to reduce bias. We were careful in this review to openly state our review approach and methodological, registering it on PROSPERO and reporting the review in accordance with PRISMA requirements. These steps are all recognised measures in the context of systematic reviews for reducing bias. 

As regards critical appraisal specifically, we chose the MetaQAT tool because it is an internationally-recognised critical appraisal tool (CAT) that has been validated including for use by non-subject area specialists. The tool focuses on broad aspects of study design and implementation. We made two minor modifications to this tool – to capture the publication type (peer-reviewed paper or pre-print) and to gather information on the assay or laboratory technique used (an area of particular relevance to the subject matter) – assessed by reference to material reported in the paper. This validated MetaQAT tool was the basis on which critical appraisals were made. 

In light of the concerns raised regarding quality appraisal, we have removed the follow-on step that we introduced when synthesising MetaQAT results for the paper: the use of a traffic-light style, low/medium/high quality categorisation (see changes tracked throughout the paper). We have instead retained our narrative comments on study quality (in both the main body of the paper and the supplementary materials). Our overall judgement remains, however, that a majority of the studies included in the review had significant methodological shortcomings that limit the aggregate conclusions that can be made from the published evidence. 

Finally, in regard to the question of preprint and peer-reviewed versions of the same studies; we included the most recent version of the relevant study that had been published within the time-frame of our searches. For reasons of rigour in the systematic review process, it was necessary to adhere to the time limits imposed by the searches and not cite later versions of specific papers published after the searches were completed as the content and even results presented may have changed post peer review. 

COMMENT: Table 2 is incomplete at best and clearly indicates that the authors have not read in detail the articles they reference. I started completing the table to indicate where the right reference should go but it is not the role of this reviewer to perform such task. This table should be completed once the authors read the references in detail (not only the abstracts).

RESPONSE: We thank the review for flagging concerns regarding table 2. We have revisited this table to verify that referencing was correct and made some modifications as outlined in tracked changes. We have not, however, been able to identify errors of fact in our assessment of the evidence presented in papers cited in this table. 

Reviewer #2

COMMENT: In Figure 3, please indicate which antigen/s were used for the kinetic prediction. If aggregate data was used from multiple antigens, be sure to clearly describe in the text and figure legend.

RESPONSE: Thank you for this comment. Figure 3 is in fact a schematic – i.e. intended to illustrate in general terms the trend over time in specific antibodies (across multiple antigens). We have included text to clarify this in the legend for the figure. Reporting of antigen type was in general very variable across included studies. Figure 2 captures this for first appearance of individual antibodies in those few studies that reported it in detail.

We hope these responses help in addressing the reviewers’ concerns. If you have any further queries please do not hesitate to contact us. 

Yours,

Sharif Ismail

ST4 Public Health Registrar

Wellcome Trust Clinical Research Training Fellow

On behalf of the study authors

---

## [Editor Report · Decision Letter 1]

4 Dec 2020

Antibody response to SARS-CoV-2 infection in humans: a systematic review

PONE-D-20-27927R1

Dear Dr. Ismail,

We’re pleased to inform you that your manuscript has been judged scientifically suitable for publication and will be formally accepted for publication once it meets all outstanding technical requirements.

Kind regards,

Nicholas J Mantis

Academic Editor

PLOS ONE
---

## [Editor Report · Acceptance letter]

23 Dec 2020

PONE-D-20-27927R1 

Antibody response to SARS-CoV-2 infection in humans: a systematic review 

Dear Dr. Ismail:

I'm pleased to inform you that your manuscript has been deemed suitable for publication in PLOS ONE. Congratulations! Your manuscript is now with our production department. 

Kind regards, 

on behalf of

Dr. Nicholas J Mantis 

Academic Editor

PLOS ONE